# Reoxygenation Modulates the Adverse Effects of Hypoxia on Wound Repair

**DOI:** 10.3390/ijms232415832

**Published:** 2022-12-13

**Authors:** Que Bai, Qian Gao, Fangfang Hu, Caiyun Zheng, Na Sun, Wenting Chen, Jinxi Liu, Yanni Zhang, Xianglong Wu, Tingli Lu

**Affiliations:** Key Laboratory of Space Bioscience and Biotechnology, School of Life Sciences, Northwestern Polytechnical University, Xi’an 710072, China

**Keywords:** hypoxia, burn, wound healing, ROS, reoxygenation

## Abstract

Hypoxia is a major stressor and a prominent feature of pathological conditions, such as bacterial infections, inflammation, wounds, and cardiovascular defects. In this study, we investigated whether reoxygenation has a protective effect against hypoxia-induced acute injury and burn using the C57BL/6 mouse model. C57BL/6 mice were exposed to hypoxia and treated with both acute and burn injuries and were in hypoxia until wound healing. Next, C57BL/6 mice were exposed to hypoxia for three days and then transferred to normoxic conditions for reoxygenation until wound healing. Finally, skin wound tissue was collected to analyze healing-related markers, such as inflammation, vascularization, and collagen. Hypoxia significantly increased inflammatory cell infiltration and decreased vascular and collagen production, and reoxygenation notably attenuated hypoxia-induced infiltration of inflammatory cells, upregulation of pro-inflammatory cytokine levels (IL-6 and TNF-α) in the wound, and remission of inflammation in the wound. Immunofluorescence analysis showed that reoxygenation increased the expression of the angiogenic factor α-SMA and decreased ROS expression in burn tissues compared to hypoxia-treated animals. Moreover, further analysis by qPCR showed that reoxygenation could alleviate the expression of hypoxic-induced inflammatory markers (IL-6 and TNF), increase angiogenesis (SMA) and collagen synthesis (Col I), and thus promote wound healing. It is suggested that oxygen can be further evaluated in combination with oxygen-releasing materials as a supplementary therapy for patients with chronic hypoxic wounds.

## 1. Introduction

Oxygen is an essential molecule that maintains cell vitality, growth, differentiation, metabolism, and even cell communication, and is crucial for the survival, function, and fate of cells [1,2,3,4]. For example, in hypoxic regions far from blood vessels, hematopoietic stem cells maintain their undifferentiated state and hypoxia influences their proliferation and cell fate [5]. Inadequate oxygen supply prevents cell migration and neovascularization, reducing cell growth and differentiation [6]. As the body’s first line of defense, the skin plays a vital role in preventing dehydration of cells and protecting internal organs from external environment damage [7]. However, when the skin suffers serious damage, including surgery and burns, it loses its necessary protective mechanism and eventually develops a wound [8]. In general, immediately after the acute injury, the damaged vascular system obstructs oxygen delivery to the wound, creating a hypoxic environment around the wound, which potently alters cellular signaling pathways for cell function. For example, the expression of multiple genes (including angiogenesis, cell migration, inflammation) changes along with the activity of hypoxia-inducible factor 1 and nuclear factor kappa B [9]. Therefore, acute hypoxia could promote cell proliferation, form a fully functional vascular network, provide adequate oxygen, and facilitate tissue repair. But in chronic injuries, such as burns, the vascular network is so severely damaged that it does not provide sufficient oxygen for extracellular matrix (ECM) synthesis and re-epithelialization, jeopardizing the healing process [10,11]. Although long-term hypoxia results in delayed wound healing, the mechanism of the reduction of environmental oxygen level affecting the wound healing process is still incomplete.

Wound healing can be roughly divided into four successive overlapping stages: haemostasis; inflammatory, proliferative, and remodeled. These highly regulated cellular molecular interactions can lead to regeneration [12,13]. After injury, components in the ECM bind and activate circulating platelets, which then undergo adhesion and aggregation, triggering the extrinsic and intrinsic coagulation pathways that together to stabilize fibrin-platelet clots [12]. Inflammation arises immediately after tissue injury and requires clotting cascades, inflammatory pathways, and immune system to prevent ongoing loss of blood and body fluids, to remove dead and inactivated (dying) tissue, and to prevent infection [12]. Then, in response to complement activation, platelet degranulation, and bacterial degradation products, neutrophils are recruited into the wound to enhance the inflammatory process [14]. The proliferative phase is characterized by cell proliferation and migration of different cell types, including the migration of keratinocytes over the damaged dermis (the inner layer of the skin) and the formation of new blood vessels (angiogenesis). Finally, the wound scar produces more collagen and elastin, and the fibroblasts mature into myofibroblasts [15]. Under certain conditions of stimulation (e.g., chronic diseases, vascular insufficiency, diabetes, and hypoxia), chronic wounds can develop, characterized by massive neutrophil infiltration, and their associated reactive oxygen species (ROS) and destructive enzymes that perpetuate circulation and exacerbate the inflammatory state leading to failure to heal [16,17,18].

Hypoxia can induce inflammation, and inflamed lesions often aggravate hypoxia, leading to a vicious cycle [19]. Hypoxia is a common feature of the microenvironment of both physiological and pathological immune ecological niches [20]. Exposure to environmental hypoxia triggers acute inflammatory responses in different organs, including the lungs, the intestine, the kidney, or the heart [21]. The inflammatory response is initiated by the aggregation of leukocytes to the site of infection/injury and this migration process occurs through a series of highly coordinated intercellular interaction steps mediated by cell adhesion molecules and integrins [22]. This increased metabolic activity exceeds the availability of oxygen, leading to inflammation that produces hypoxic conditions locally. Hypoxia at the site of inflammation can lead to vascular rupture, coagulation, obstruction of blood flow, and accumulation of inflammatory cells in hypoxic organs or release of inflammatory mediators (e.g., tumor necrosis factor [TNF], interleukin [IL]-6 or IL-8). Tissue necrosis will occur with the long-term loss of oxygen delivery [23]. Thus, understanding the regulation of hypoxia-mediated pathways in inflammation, and the resolution of this process, could provide the basis for tissue damage therapies.

One study showed that hypoxia induces lipid and protein oxidation, while reducing the antioxidant defense system in the brain [24]. After reoxygenation, oxidative stress parameters and antioxidant system are returned to the control levels. However, severe hypoxia can increase the oxidative damage of cells, thus activating the apoptosis mechanism [25,26,27]. In addition, Terraneo et al. showed that the prolonged mild hyperoxia led to sustained brain damage, which was similar to that suffered from prolonged mild hypoxia. The imbalance between reactive oxygen species generation and anti-ROS defense was similar, but the level of imbalance occurs at higher levels in hypoxia than in hyperoxia [28]. Although there have been many studies on the effects of hypoxia on brain tissue [29,30,31], little work has been done on tissue repair.

The aim of the present study was to evaluate the effects of hypoxic exposure on cell and wound healing. L929 and HUVECs cells, related with the wound repair, were chosen and exposed under the hypoxia or reoxygenation conditions. Moreover, we investigated whether reoxygenation has a protective effect against hypoxia-induced acute injury and burn using the C57BL/6 mouse model. C57BL/6 mice were exposed to hypoxia and treated with both acute and burn injuries and were in hypoxia until wound healing. Next C57BL/6 mice were exposed to hypoxia for three days and then transferred to normoxic conditions for reoxygenation until wound healing. Finally, skin wound tissue was collected to analyze healing-related markers, such as inflammation, vascularization, and collagen. The results might provide some enlightenment concerning wound repair under hypoxia, as well as references for the treatment of the chronic wounds with the similar hypoxia condition.

## 2. Results

### 2.1. Cell Proliferation Inhibited in Hypoxia

Hypoxia is a common phenomenon in a range of diseases that can significantly affect the microenvironment of the lesion. In order to explore the effect of hypoxia on cell growth, the CCK-8 test was pe to study the proliferation of L929 and HUVECs cells. When L929 and HUVECs were cultured at 1% (hypoxia) and 21% (normoxia) oxygen concentrations for 1, 2, 3 and 4 days, cells in normoxia exhibited higher proliferation rate than that in hypoxia (Figure 1B,C). Compared to cells cultured in normoxia, the viability of L929 in hypoxia was not obviously different at 1 day, but was reduced at 2 days and only half of that in normoxia at 3 days, as well as an evident decrease at 4 days (Figure 1B). These results indicated the significant inhibitory effect of hypoxia on L929 proliferation. For HUVECs, compared with that in normoxia, there was no evident difference in cell proliferation in the first two days under hypoxia, but it was visibly reduced at 3 days and 4 days (Figure 1C). Besides, cell proliferation was further analyzed using a CaM/PI kit. In Figure 1D,E the activity of cultured cells under hypoxia was much lower than that in normoxic group. The above results of CaM/PI were consistent with CCK-8 assay. In conclusion, hypoxia (1%) showed a significant inhibitory effect on the cell proliferation of both L929 and HUVECs.

### 2.2. ROS and MMP Changed in Hypoxia

Redox balance is critical to cell physiology, and the increase of intracellular ROS production can lead to oxidative stress damage to cells. So, the changes of intracellular ROS can be detected by a ROS indicator DCFH-DA (2′,7′-Dichlorofluorescin diacetate), and its staining was used to assess the ROS levels of cells responding to the hypoxia stimulation. DCFH-DA has no fluorescence and can freely cross the cell membrane. After entering cells, DCFH-DA is hydrolyzed by intracellular esterase to generate DCFH, and subsequently oxidized into fluorescent compound DCF (2′,7′-dichlorofluorescein) by intracellular ROS, such as H_2_O_2_ and OH^●^. The level of intracellular ROS can then be determined by fluorescence spectroscopy [32]. Compared with the ROS-up positive control group, the green fluorescence of L929 and HUVECs cells was significantly lower than that of normoxia, while their fluorescence was comparable to that of hypoxia, suggesting that in-tricellular oxidative stress is effectively enhanced under hypoxia (Figure 2A,B). Moreover, the mean fluorescence density of ROS also indicated that hypoxia promoted the production of ROS (Figure 2C,D).

Mitochondria are important to the cell energy metabolism and apoptosis regulation, as well as indispensable for cell survival and growth. Mitochondrial membrane potential (MMP) can reflect the health state of mitochondria. Therefore, the mitochondrial biogenesis and membrane potential level were investigated under hypoxia. A mitochondrial membrane potential assay kit with JC-1 (Beyotime Biotechnology, China) was used to test the changes of cells membrane potential. JC-1 gathers in the matrix of mitochondria and forms a polymer (J-aggregates), which produces red fluorescence when MMP is at the high level. While JC-1 is monomer and can produce green fluorescence with low level of MMP. Furthermore, the decrease of MMP is a landmark event in the early stage of apoptosis [33]. The results showed that the mitochondria, both L929 and HUVECs, showed the red fluorescence after culture for 24 h in normoxia, with the very weak green fluorescence. However, mitochondria of cells in hypoxia presented a reduced MMP and a strong green fluorescence (Figure 3A,C), indicating cells were in the early stage of apoptosis. Besides, the corresponding fluorescence intensities were further quantified using ImageJ software (Figure 3B,D), which was consistent well with the results shown in Figure 3A,C. Moreover, the effect of hypoxic culture on cell apoptosis was detected using the Annexin V/PI kit, as shown in Figure 3E. After 24 h of cell culture, almost no green and red fluorescence was observed under normoxia, while the hypoxic group showed strong green and red fluorescence, indicating that L929 cells were in a state of apoptosis/necrosis. It was further shown that hypoxia was detrimental to cell growth and survival.

### 2.3. mRNA Expression of Collagen and Anti-Inflammatory Factors Decreased in Hypoxia

Collagen is the natural substrate for cell attachment, growth, and differentiation, promoting cell proliferation and differentiation [34]. During the process of wound remodeling, the acellular matrix actively remodeled from a predominantly type III collagen skeleton to a mainly type I collagen, thus forming mature ECM, improving the epidermis and increasing the tensile strength [12]. Therefore, in this experiment, we detected the effect of hypoxia on collagen. So, the mRNA expressions of key collagen proteins including Col I, Col III and fibronectin were investigated. The mRNA levels of Col I, Col III and fibronectin of L929 decreased when cells were cultured in hypoxia for 12 h and 24 h (Figure 4A–C). It was found that hypoxic microenvironment may attenuated collagen synthesis in L929 cells. Excessive deposition of collagen type I is one of the hallmarks of healing-related fibrosis. The results suggest that the decrease of Col I mRNA upon hypoxia might also indicate non-fibrotic outcome of healing. Similarly, regarding the Col III mRNA level, as collagen type III is widely recognized as a marker of scarless, regenerative healing, the results might suggest that normoxia is superior to hypoxia in terms of pro-regenerative III collagen deposition.

Oxygen supply is essential to the immune response because every cell in the immune system must perform its function in an environment with a specific level of oxygenation [35,36,37]. As immune cells move through the wound tissue, the availability of oxygen changes rapidly, adapting to the needs of the cells [36]. Studies have shown that hypoxia can activate immune cells, such as neutrophils, macrophages, and lymphocytes, and thus increase energy requirements, resulting in hypoxic microenvironment and mitochondrial dysfunction. Therefore, RAW264.7 cells were cultured under normoxia and hypoxia. The expression of proinflammatory M1 macrophages and anti-inflammatory M2 macrophages marker genes were analyzed, respectively. As shown in Figure 4D–F, after 12 h of culture, there was no significant difference in the expression of IL-6 and TNF-α genes between normoxia and hypoxia groups, while the expression of IL-1β was significantly up-regulated (Figure 4D–F). After 24 h, the expression of IL-1β showed no significant difference, while IL-6 and TNF-α were significantly up-regulated. In hypoxia group, pro-inflammatory factors were up-regulated while anti-inflammatory factors were down-regulated, which may break the balance between the pro-inflammatory and anti-inflammatory factors leading to inflammation disorder and is not conducive to wound repair. For M2 marker genes, IL-10, Arg-1, and TGF-β did not change significantly after 12 h in hypoxia, but all of them were distinctly down-regulated after 24 h under hypoxia (Figure 4G–I). These results supported that hypoxia could inhibit the secretion of M2-related factors in macrophages. In conclusion, the results suggested that hypoxia was not only detrimental to cellular collagen synthesis, but also inhibited M2 macrophage polarization, and would delay the wound healing in the end.

### 2.4. Acute Wound Healing under Hypoxia In Vivo

The animal experiment was applied to study the effect of hypoxia on wound healing. In order to avoid hypobaropathy caused by the rapid decline of partial oxygen pressure, we gradually reduced the proportion of oxygen intake to 10% at a rate of 2% for 5 days. At this time, full-thickness skin wounds were constructed on the back of mice, which were then exposed to different oxygen conditions until the wounds healed (Figure 5A). During the various oxygen treatments, all mice survived with no significant changes in their body weight (Appendix A). As shown in Figure 5B,C, the wound closure of mice in normoxia group was highly superior to that in hypoxic group during the whole period. The wound area on day 3 was larger in mice in the hypoxic group than in the normoxic group, indicating that hypoxia had a detrimental effect on wound closure. At this time, half of mice were randomly removed and exposed to the normal oxygen condition, called reoxygenation, to test the effect of oxygen recover in the wounds. On day 5, all wound areas were significantly reduced, especially in normoxia and reoxygenation groups, showing the faster wound closure speed than that in hypoxia (Figure 5D). For instance, the wound healing rate in the normoxia and reoxygenation groups was approximately 63%, while it was only 49% in the hypoxic group. The difference in wound healing among three groups remained until the wound healed. On day 11, the wound healing rate was about 93% in hypoxic group, showing an incomplete healed. Meanwhile, the wounds in the normoxia and reoxygenation were healed completely and covered with hair. Compared with natural healing time of 11 days in normoxia, the prolonged healing period of acute mice wounds was approximately 13 days in hypoxia. So, the delayed wound healing was obvious under hypoxia, which was two days later than that of in normal or reoxygenation. This finding confirmed the important function of oxygen for wound healing and that the low oxygen concentration (such as hypoxia) could postpone the process of wound healing.

### 2.5. Histological Evaluation of the Regenerated Tissue in Acute Wound

Haematoxylin and eosin (H&E) and Masson trichrome staining were used to investigate the inflammatory response, collagen fiber remodeling and regenerative skin tissue during wound healing. The H&E and Masson staining results of the normal skin tissues were shown in Appendix A. H&E staining results of three research groups at different time were showed in Figure 6A. For three days, the wound closure of mice in hypoxia had obvious defects, with considerable inflammatory cells infiltrating into the wound surface (Figure 6A), while there were fewer inflammatory cells in the wounds of normoxia group (Figure 6A). For seven days, functional skin reconstruction of wounds, such as epidermis, dermis, and granulation tissue, were all appeared in the normoxia and reoxygenation group. However, the wound in hypoxia was still in the phase of epithelial regeneration with defects in the dermis (Figure 6A,B). On day 11, complete epithelial and dermal structures were formed in each group. The mice wound in the normoxia group developed the normal remodeled tissue with a high number of skin appendages (such as numerous mature hair follicles) (Figure 6A,C,E). For the wounds in the hypoxic group, the immature thicker epithelium was transformed into mature thinner epithelium, with significantly fewer hair follicles than in the normoxic group (Figure 6A). From the Masson staining results in Figure 6B, one can observe less collagen deposition and a looser arrangement under hypoxia. Furthermore, it was also found that hypoxia group showed the smallest area of mature tissue among three groups, as well as fewer hair follicles and blood vessels than the normoxia group (Figure 6D,E).

### 2.6. Burn Wound Healing under Hypoxia In Vivo

To evaluate the effect of hypoxia on burn wounds, a deep second-degree burn wound model was employed, and no drug intervention was performed during self-healing under different conditions. Representative photographs and schematic diagram of burn wounds at different test day were shown in Figure 7A,B. On day 0, whitening symptoms, as well as the redness and swelling of surrounding tissues, were found in all burn wounds. As shown in Figure 7A, the wound area in all groups gradually became smaller with the time increase, which was agree with the results of wound healing rates (Figure 7B,C). Along the same timeline, the wound healing rate in hypoxia was always lower than that of in normoxia (Figure 7C). Take the wound healing rate, for instance, it was about 44% in normoxia or reoxygenation on 5 days, while it was about 30% under hypoxia. On day 7, the healing rate of scald wound in normoxia and reoxygenation was significantly increased to about 57%, but was approximately 37% in hypoxia. The completely healed burn wound could be obtained in the normoxia after 13 days, showing closure two days late. Therefore, the delayed burn wounds healing was found under hypoxic conditions, which was similar to the acute wound healing under the same conditions.

### 2.7. Histological Evaluation of Burn

The effect of hypoxia on burn healing was further evaluated on the pathological level. The H&E staining histological assessment of burn wound healing at different days was demonstrated in Figure 8A. At day 3, all burn wounds presented the attenuated epidermis with the coagulative damage to the deep partial-thickness of the dermis. On day 7, inflammatory cell infiltration increased under hypoxic conditions. In contrast, inflammatory cells at the wound sites were obviously reduced under normoxic and reoxygenation (Figure 8A). On day 13, each group was completely epithelialized, and inflammation around the wound surface disappeared. Skin appendages and dermal tissue were regenerated in normoxic group (Figure 8A). While in the hypoxic group, the basic structure of epithelium and dermis was regenerated, but the hair follicles were not formed in the center of the traumatic tissue, indicating incomplete burn wound healing (Figure 8A,C). The formed collagen fibers were estimated with Masson’s trichrome staining (Figure 8B). On day 7, a large amount of ordered collagen fibrils were observed in normoxia and reoxygenated groups, while only a small number of collagen fibrils were observed in hypoxia. On day 13, the thickness of granulation tissue in hypoxia was thinner than that in normoxia or reoxygenation groups (Figure 8B,D). These results suggested that regenerated skin tissue under hypoxia shows fewer new granulation tissues and hair follicles, which is not conducive to burn repair.

### 2.8. Gene Expression Analysis

Inflammation is a nonspecific, innate immune response that involves the breakdown of tissues and the removal of cellular, extracellular and pathogen debris [18]. Chronic wounds are usually in an inflammatory state, preventing proliferation. It is well known that local tissue hypoxia can severely disrupt wound healing and promote inflammatory cascades. The significant expression of endothelial adhesion molecules in hypoxic tissues contributes to the subsequent extravasation of neutrophils and macrophages. Recruited neutrophils and macrophages then synthesize pro-inflammatory cytokines, such as IL-1α, IL-1β, IL-6, and TNF-α, in an autocrine manner, which perturbs the balance between proteases and their inhibitors, perpetuating inflammation. The result of hypoxia is an increased inflammatory state that hinders healing. In this study, we mainly examined the expression of the pro-inflammatory cytokines TNF-α and IL-6. At three days, in acute wounds, TNF-α gene expression was significantly upregulated in hypoxic group compared with normal skin. TNF-α expression was still at a high level in the hypoxic group on days 7 and 11 (compared with the normoxic group) (Figure 9A). The expression of IL-6 was not obvious in the hypoxic group at 3 d (compared with normal skin). At day 7, the gene expression was up-regulated in hypoxia group, while IL-6 gene was significantly down-regulated in reoxygenation group (compared with normal skin) (Figure 9B). These results indicate that prolonged hypoxia could activate the high expression of proinflammatory factors at the wound site, which is obviously alleviated by reoxygenation. In burns, TNF-α gene expression was upregulated (compared with normal skin) in normoxic and hypoxic groups at 3 d (Figure 9F), indicating that inflammation was more pronounced and prolonged in burns than in acute wounds. At seven days, the TNF-α gene was significantly upregulated in hypoxia group, but there was no significant difference in the other groups. On day 3, IL-6 gene expression was increased in the hypoxia group (compared with normal and normoxia), and was down-regulated in the normoxia group on day 7. Although the expression of IL-6 gene was not different in hypoxia group compared with normal skin, it was up-regulated compared with normoxia group, and reoxygenation alleviated the proinflammatory state of IL-6 (Figure 9G).

Angiogenesis is a key indicator for evaluating wound repair because enhanced angiogenesis could provide oxygen and nutrients to cells and accelerates the cell migration, thereby result in promoting collagen synthesis, granulation tissue formation, and wound healing [38,39]. Therefore, angiogenesis of the wounds was assessed by expression of vascular related genes, such as CD31 and α-SMA. As shown in Figure 9C, there was no significant difference in α-SMA expression in acute wounds on day 7. At 11 d (late wound healing stage), it was significantly upregulated in all three groups compared to normal skin, indicating that angiogenesis was underway at this stage. α-SMA expression was decreased in hypoxic group compared to normoxic group, while it was significantly increased in the reoxygenated group. In addition, CD31 gene expression was down-regulated in the hypoxic group (compared with normal oxygen) at 11 d, while reoxygenation could effectively improve CD31 expression (Figure 9D). In burns, α-SMA expression was highly significantly increased in the normoxia group at 13 d, while the hypoxic group showed a negligible increase, and the reoxygenation group also showed an increasing trend (Figure 9H), indicating that long-term hypoxia could inhibit angiogenesis at the wound, while reoxygenation could effectively increase angiogenesis and promote wound healing.

Formation of collagen in wound sites is a very important sign for the remodeling of damaged skin tissue [40]. Thus, qPCR was employed to further examine the expression of collagen at the wound site. As shown in Figure 9E, acute wounds were not epithelialized at day 3, so there was no difference in Col I expression. Compared with normal skin on day 7, Col I expression was up-regulated in all three groups, with the lowest expression in hypoxic group. The gene expression of 11 d and 7 d was similar, in which Col I expression was lower in hypoxic group, so the wound collagen reconstruction was slower, which was consistent with Masson (Figure 6B). In burns, due to severe wound injury, Col I expression showed no difference between three and seven days. Compared with normal skin, Col I expression was significantly increased in all three groups at day 13, among which hypoxic group had the lowest expression (Figure 9I). It can be concluded that hypoxia restricts collagen synthesis, while reoxygenation could effectively increase collagen reconstruction.

### 2.9. Immunohistochemical Analysis

In addition, angiogenesis of the wounds was assessed by the immunofluorescent staining for CD31 and α-SMA (α-smooth muscle actin). Platelet endothelial cell adhesion molecule-1 (CD31 or PECAM-1) is a transmembrane glycoprotein that belongs to the immunoglobulin superfamily [41]. CD31 has been implicated in the adhesive and signaling events required for endothelial cell migration, a process critical to angiogenesis. α-SMA is a sign that cells are derived from vascular smooth muscle tissue [42]. It is commonly found in mature vascular smooth muscle cells and perivascular cells/pericytes around small blood vessels. As a marker protein of endothelial progenitor cells and vascular smooth muscle cells, α-SMA plays an important role in the process of vascular regeneration. For acute wounds in hypoxia, the results showed that the significantly lower relative coverage of CD31/α-SMA-positive than that of normoxia (Figure 10A,E,F). Besides, for reoxygenation, the results were similar to that in normal oxygen condition (Figure 10A), indicating that reoxygenation promoted angiogenesis. For burns, the results showed the significantly lower relative coverage of α-SMA-positive in hypoxia than that in normal oxygen (Figure 10B,G), indicating that the inhibited neovascularization was exposed to hypoxia, which was not conducive to oxygen exchange and nutrient supply. Moreover, the relative coverage of α-SMA-positive in reoxygenation were significantly higher than that in hypoxia (Figure 10B), which was consistent with qPCR results (Figure 9E).

Oxidative stress is a main factor hindering wound recovery [43], so we speculated that the slow wound repair under hypoxia may be caused by oxidative stress. So, the ROS levels in wound site were evaluated by DHE (dihydroethidium) staining [44]. For the acute wound, the staining results showed that there was no significant difference in red fluorescence among the three groups (Figure 10C), indicating that ROS homeostasis was not broken in acute wounds. Meanwhile, for the burn wound, the peroxidation products were detected in the hypoxia lesion sites and the dramatically increased ROS level was found (Figure 10D). However, reoxygenation significantly reduced ROS production compared to the hypoxia following for 13 days (Figure 10D). These results indicated that hypoxia does not break the ROS homeostasis of acute wounds, while it could induce the increase of ROS in burn wounds, and thus delay the healing of burn wounds.

## 3. Discussion

Hypoxia and inflammation are common incidental events in various pathological immune processes, including chronic inflammation and ischemic tissue [20]. The effect of hypoxia on inflammation is mainly driven by HIF-1/2α [45]. HIF-1α and HIF-2α regulate variety of cellular signaling molecules involved in the inflammatory response and can also regulate multi-various immune processes, such as M1/M2 polarization [46]. Hif-1α regulates NF-κB signaling by modulating IκB (NF-κB inhibitor) (Appendix A). The interaction between HIFs and NF-κB plays an important role in the hypoxic immune response [47]. Nuclear factor κB (NF-κB) signaling is activated by both inflammatory stimuli (such as cytokines or bacterial products) and hypoxia. Under hypoxic conditions, it was found that increased activity of the inhibitor of κB kinase (IKKβ) and phosphorylation of the κBα inhibitor (IκBα) result in the translocation of NF-κB subunits (p65 and p50) to the nucleus, thereby activating transcription of inflammatory genes, such as TNF-α, IL-1β, VCAM, and IL-8 [20]. In our study, H&E results showed that reoxygenation treatment reduced inflammatory cell infiltration. In addition, qPCR data showed that reoxygenation significantly reduced the expression levels of IL-6 and TNF-α, and effectively alleviated the early inflammatory response of the wound under hypoxia. Therefore, oxygen-carrying materials can be applied in the future to treat inflammatory caused by chronic hypoxic wounds.

It has been widely believed that hypoxia and inflammation are closely related to oxidative stress [48]. ROS is produced by mitochondrial complex III and L-2-hydroxyglutaric acid (L-2HG) under hypoxia, which are the main source of mitochondrial genomic instability and lead to respiratory chain dysfunction [9,49]. The cells’ mitochondrial membrane potential was decreased after 24 h cultured in hypoxia, indicating the cells were damaged. Impaired mitochondrial function and oxidative damage caused by hypoxia further exacerbated inflammatory responses through metabolic disturbances [49]. The result was in agreement with the statement that ROS in excessive inflammation response could aggravate local tissue damage and lead to chronic inflammation [32,50].

ROS is the key signaling molecules involved in cell proliferation, apoptosis, and cell homeostasis that attack and destroy the immune system of bacteria [51,52]. However, excessive ROS production can cause the metabolic damage of DNA, proteins, lipids, and carbohydrates, resulting in impairment of cellular function and wound healing or regeneration of damaged tissues [53,54,55]. It has been documented that if mouse fibroblasts are exposed to excessive chronic oxidative stress, their replicative lifespan will be affected and they will become senescent prematurely under hypoxia conditions [56]. Oxidative stress induces the excessive production of ROS and reduces the function of the antioxidant systems [57]. Moreover, there is increasing evidence that activation of HIF-1α during hypoxia is associated with oxidative stress and directly or indirectly regulates ROS formation [58]. Under hypoxic conditions, HIF-1α is stabilized and ROS formation is increased [29]. Increased ROS levels contribute to further activation of HIF-1α. Here, hypoxia would induce oxidative stress. In our study, immunofluorescence and qPCR data showed that hypoxia exacerbated intracellular ROS and inflammatory gene expression. Animal experiments further showed that hypoxia promoted ROS production and inflammatory factor (TNF-α and IL-6) expression, while reoxygenation treatment significantly reduced ROS levels, which may be associated with reduced wound inflammation. These findings suggest that reoxygenation may act as an anti-inflammatory agent, protecting the wound from excessive inflammation harm, and without any adverse effects, such as increased risk of infection. However, we recommend that additional studies in this area to help assess the clinical safety in infection models.

Impaired angiogenesis often delays wound healing since neovascularization is necessary for adequate oxygen exchange and nutrition to wounds, and it also considered as one of the key events in wound healing [44]. Enhanced angiogenesis can accelerate the cell migration to the wound site, and thus promote collagen synthesis, granulation tissue formation and wound healing [39]. Since α-SMA is related to actin and integrin, α-SMA myofibroblasts are mediated by β1 integrin, and its high expression facilitates the migration and differentiation of fibroblasts to myofibroblasts, thus promoting granulation tissue formation, and promote re-epithelialization and angiogenesis [38,59]. Previous studies have shown that genetic depletion of αSMA myofibroblasts leads to multipotent wound healing defects, including a lack of re-epithelialization and granulation, impaired angiogenesis, and increased hypoxia, which are hallmarks of chronic non-healing wounds [59,60]. Our study demonstrates that reoxygenation treatment is essential for wound healing by promoting α-SMA expression, which in turn promotes wound contractility, re-epithelialization, and angiogenesis.

Blood vessels transport oxygen, nutrients, and wastes, and can also act as pathways for cellular signaling molecules, and are involved in determining the fate of progenitor cells [61,62]. Therefore, it is very important to study the influence of blood vessels on tissue regeneration. CD31 is known as platelet endothelial cell adhesion molecule-1 (PECAM-1/CD31) and plays a vital role in cell migration, angiogenesis, and the activation of integrin [63]. The results of this study suggested that reoxygenation could stimulate CD31 angiogenesis and promote skin regeneration in mice. Together with the above results, it was speculated that hypoxia could inhibit angiogenesis at the wound site, resulting in delayed healing. Moreover, reoxygenation can effectively increase vascular gene expression, thus activating cell differentiation and migration to the wound and promoting wound repair. Collectively, these results suggested that appropriate oxygen was critical factor for blood vessel re-building.

The quality and composition of the dermal ECM is critical to success of wound healing [64,65]. The formation of a healthy ECM contributes to normal mechanical and cellular properties. Our wound morphology Masson’s trichrome staining analysis showed that hypoxia-treated wounds resulted in sparse epidermal coverage, thin epidermal layers, and weak collagen staining. These findings are typical of less mature epidermis, suggesting that it develops more slowly than reoxygenated skin-treated wounds. In contrast, reoxygenated skin-treated wounds have thicker epidermis and granulation tissue. Their regenerated wounds were able to heal faster and had a mature and healthy epidermal coverage.

By reducing inflammation, increasing angiogenesis, and maintaining collagen deposition, reoxygenation therapy may improve hypoxia-induced chronic wound healing. In the clinical setting, given the pleiotropic effects of reoxygenation therapy, reoxygenation may potentially block the inflammatory cascade and ultimately provide a new therapeutic option for accelerating tissue damage repair. Future studies will involve in-depth characterization of immune cells at the site of inflammation, particularly macrophages. Gene expression profiling techniques such as RNA-Seq are needed to characterize the functional properties of cells at the site of inflammation. Further understanding of the mechanisms of reoxygenation for tissue damage is critical for the development of clinical therapeutic interventions, particularly in patients with chronic hypoxic injury.

## 4. Materials and Methods

### 4.1. Materials

Dichloromethane (99.5%) was purchased from Haohua Chemical Reagent Co. Ltd. (Luoyang, China). Lsopropyl alcohol (99.7%) was obtained from Tianli Chemical Reagent Co. Ltd. (Tianjin, China). DNase/RNase-Free Water (DEPC water) and Calcein-AM/PI double stain kit were obtained from Solarbio Science & Technology Co., Ltd. (Beijing, China). Annexin V/PI kit was purchased from Bio-Box. Cell Counting Kit-8 (CCK-8), Reactive Oxygen Species Assay Kit (ROS Assay Kit) and Mitochondrial membrane potential assay kit (JC-1) were bought from Beyotime Biotechnology. Cell culture media (DMEM), fetal bovine serum (FBS), phosphate-buffered saline (PBS), and antibiotics were obtained from Biological Industries. All chemicals were of analytical or equivalent grade and used without further treatment.

### 4.2. Hypoxia Chamber

A modular incubator chamber (Billups-Rothenberg Inc., San Diego, CA, USA) was used for cell hypoxia studies. Hypoxia modular incubator chamber with an upper sealing chamber, a lower sealing chamber (connected with inlet and outlet gas pipes), a partition frame made of polycarbonate material, and a stainless-steel sealing fastener (Figure 1A). The CO_2_ was proportionally filled into the cylinder (1% O_2_, 5% CO_2_). The gas pipe was connected to the cylinder and the device, and the gas was filled at a flow rate of 25 L/min for 4 min duration. The oxygen concentration was monitored with an O_2_ electrode (Nuvair, Oxnard, CA, USA). Note: 1% O_2_ was used for all in vitro experiments.

### 4.3. Cell Culture and Activity Assay

Mouse macrophage (RAW 264.7) and human umbilical vein endothelial cells (HUVECs) were purchased from the China Center for Type Culture Collection. Mouse fibroblast cells (L929) were obtained from the Cell Bank, School of Life Sciences, Northwestern Polytechnical University. Specific methods of cell culture are shown in the Appendix A.

The effect of hypoxia on cell viability was determined by the CCK-8 assay (Beyotime Biotechnology, China) in vitro. Take L929 as an example. Briefly, L929 cells were seeded into 48-well culture plates at the density of 2 × 10^4^ cells per well, and incubated at 37 °C in an incubator with 5% CO_2_ for 12 h. Afterwards, the cells were cultured in normal (21% O_2_ and 5% CO_2_) and hypoxia (1% O_2_ and 5% CO_2_) culture chambers. After 1, 2, 3, and 4 days incubation, cells were gently washed once with sterile PBS and then treated with 100 μL fresh culture medium (without phosphate-buffered saline) and 10 μL CCK-8 solution, and further incubated at 37 °C for 2 h. Then, the cell viability was quantified by measuring the absorbance value at 450 nm using a microplate reader (Synergy HT, Winooski, VT, USA). In addition, L929 and HUVECs cells were incubated with CaM/PI staining and observed with an inverted fluorescence microscope (MD IL HC, Wetzlar, Germany). Tests of each group were repeated three times.

### 4.4. Reactive Oxygen Species Measurement

Hypoxia could disrupt redox homeostasis, so the effect of hypoxia on cellular reactive oxygen species (ROS) was examined. L929 and HUVECs cells were inoculated into 24-well plates at a density of 1 × 10^6^ cells/well, respectively. After 12 h of adherent culture, the cells were treated under normal and hypoxia, respectively. The level of intracellular ROS was then measured using ROS assay kit (Beyotime Biotechnology, Haimen, China). Detailed operation methods are presented in the Appendix A.

### 4.5. Mitochondrial Membrane Potential (MMP) Measurement

L929 and HUVECs cells were inoculated into 24-well plates at a density of 1 × 10^6^ cells/well, respectively. After 12 h of adherent culture, the cells were separately treated under normal and hypoxia. Mitochondrial membrane potential was detected by MMP assay kit with JC-1 (Beyotime Biotechnology, China). Detailed operation methods are presented in the Appendix A.

### 4.6. Annexin V/PI Measurement

L929 cells were inoculated into 48-well plates at a density of 1 × 10^5^ cells/well. The cells were separately treated under normal and hypoxia. Apoptosis was detected by Annexin V/PI kit (Bio-Box, BA1250, Pretoria,South Africa). After 24 h incubation, the medium was aspirated and PBS washed 2 times. Then, 500 μL of binding solution, 5 μL of Annexin V, and 5 μL of PI were added, incubated for 10 min at room temperature in darkness, and observed using inverted fluorescence microscopy (MD IL HC, Leica, Wetzlar, Germany).

### 4.7. RNA Extraction and Quantitative Real-Time PCR

The total RNA from cells and skin tissue were extracted using AG RNAex Pro Reagent (Accurate Biology, Shanghai, China). cDNA was synthesized using a reverse transcription system kit according to the manufacturer’s instructions (Evo M-MLV RT Premix for qPCR, Accurate Biology, China). Quantitative real-time PCR was performed using the SYBR Green Premix Pro Taq HS qPCR kit (Accurate Biology, Shanghai, China) following the manufacturer’s protocol. PCR conditions were as follows: one step at 95 °C for 10 s, one step at 60 °C for 30 s, 72 °C for 5 s, and 2 s at 80 °C. All the above steps have 40 cycles. Gene expression levels were normalized to GAPDH and analyzed using the comparative cycle threshold (2^−ΔΔCt^) method. Primer sequences for qPCR used in this study were listed in the Appendix A.

### 4.8. Hypoxia Wound Models in Mice

Male C57BL/6J mice (6–8 week, 20 ± 0.2 g) were purchased from the Experimental Animal Center of Xi’an Jiaotong University. All mice were housed in the animal facility of the School of Life Sciences, Northwestern Polytechnical University, maintained at 22 ± 1 °C (50 ± 5% relative humidity) with 12 h light/dark cycles with drink freely and eat normally. All animal experimental protocols were approved by the Ethics Committee for the Use of Experimental Animals of Northwestern Polytechnical University (No: 202202091). All experiments complied with all relevant ethical regulations.

For acute wounds, the animals were anesthetized by pentobarbital sodium (Xianshiji, China). The back hair was scraped with electric clipper. Then, the hair was removed with hair removal cream and sterilized with 75% alcohol, and a hole punch with a diameter of 6 mm was used to form a wound on the back of each mouse (Figure 5A). Finally, a circular skin wound with full-thickness was created on the mouse.

Deep second-degree burn models were established on the dorsum of C57 mice using a preheated copper metal rod. Briefly, after mice anesthesia, hair removal, and disinfection with 75% ethanol, the 6 mm copper metal rod was heated in a water bath at 100 °C for 10 min and immediately contacted the mouse skin directly for 10 s without any external force. Each mouse developed a deep second-degree burn wound on its dorsum.

After modeling, the mice were raised in separate cages to prevent scratching each other’s wounds. The mice were separated to 3 groups with 18 in each group. The mice in normal oxygen were as the control group (normoxia). Other 36 mice were raised in the Hypoxia Chamber (FLYDWC50-IIC, Guizhou, China) with the low oxygen concentration of 10% O_2_. After 3 day, 18 mice were randomly removed from the hypoxia chamber and raised in normal oxygen again as the reoxygenation. The other 18 mice were raised in hypoxia chamber all the time as the hypoxia group. The detail of animal experiment was showed in Figure 5A. During the experiment period, all mice were free to drink water and eat normal food.

### 4.9. Histological and Immunofluorescence Staining Analysis

After 3, 7, and 11 days of treatment, the C57 mice were euthanized by injection of excessive pentobarbital sodium, and the skin tissues were removed for H&E, Masson, and immunofluorescence staining. Specifically, neovascularization CD31 and α-smooth muscle (α-SMA) were observed and quantitatively analyzed. More experimental details are provided in the Appendix A.

### 4.10. Statistical Analysis

Data were analyzed using GraphPad Prism 8 software. All experimental data were expressed as mean ± SD (standard deviation), and each experiment was performed a minimum of three times. Significant differences between groups were determined using a one-way or two-way analysis of variance (ANOVA). More experimental details are provided in the Appendix A.

## 5. Conclusions

Prolonged exposure to hypoxia may slow the process of wound repair. Hypoxia-induced injury impairs angiogenesis and collagen deposition, increases inflammation, and has profound effects on mitochondrial homeostasis (Figure 11). Although many studies have attempted to uncover the causes of wound healing, the molecular drivers of hypoxic injury still need to be investigated. On the basis of the aerobic properties of wound repair, we hypothesized that exogenous reoxygenation could protect against the adverse effects of hypoxia injury by inhibiting inflammatory mediators, reducing oxidative stress, restoring vascular function and increasing collagen deposition (Figure 11). Our work confirmed that reoxygenation significantly increases the progression of wound healing in mice. Therefore, reoxygenation can effectively alleviate the adverse effects of hypoxia, so it can be used as a strategy for the subsequent treatment of hypoxic wounds.

## Figures and Tables

**Figure 1 ijms-23-15832-f001:**
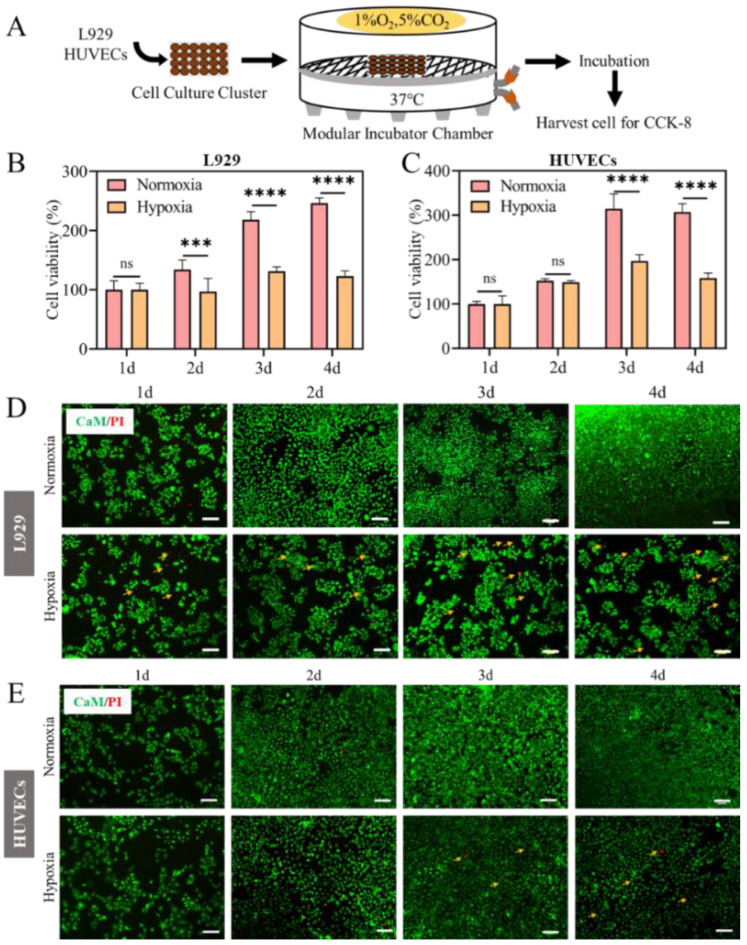
Effect of hypoxia on cell proliferation of L929 and HUVECs. (**A**) Diagram of cell culture in hypoxia (1% O_2_) and normoxia (21% O_2_) for different culture time. Cell viability of L929 (**B**) and HUVECs (**C**). Live-dead staining of L929 (**D**) and HUVECs (**E**). The yellow arrow represents dead cells. (Calcein AM staining showed green fluorescence in living cells. While Propidium Iodide (PI) stained dead cells and showed red fluorescence). Scale bar: 200 μm. (Data are presented as mean ± S.D. *n = 6*, n.s. no significance, * indicates significant difference, *** *p* < 0.001, **** *p* < 0.001).

**Figure 2 ijms-23-15832-f002:**
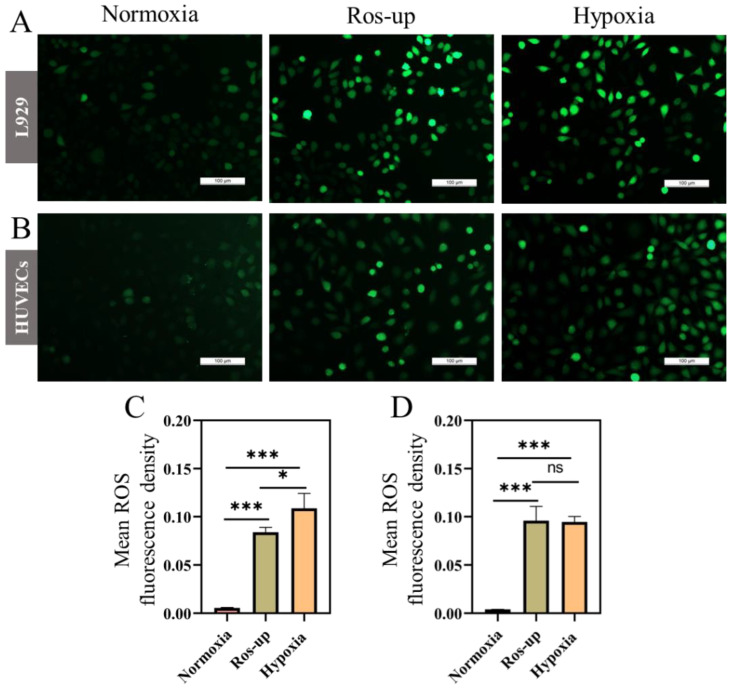
Effects of hypoxia on cellular ROS level. Representative ROS staining (green fluorescence) of L929 (**A**) and HUVECs (**B**) under hypoxia (1% O_2_) and normoxia (21% O_2_) after 24 h. The mean ROS fluorescence intensity of L929 (**C**) and HUVECs (**D**) on various treatment. Scale bar: 100 μm. (Data are presented as mean ± S.D. *n = 3*, n.s. no significance, * indicates significant difference, * *p* < 0.05, *** *p* < 0.001).

**Figure 3 ijms-23-15832-f003:**
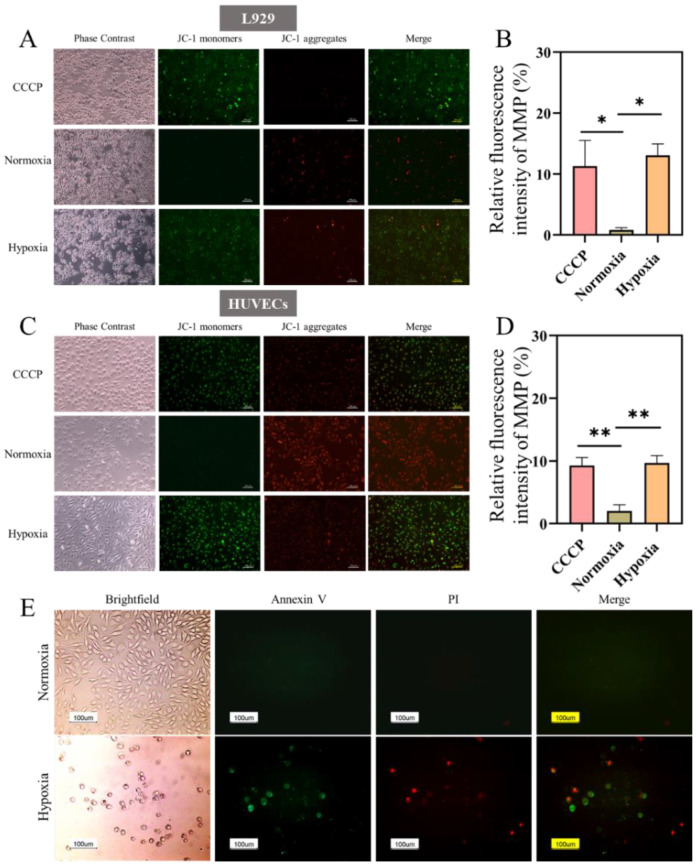
Mitochondrial membrane potential of L929 (**A**) and HUVECs (**C**) cultured under hypoxia (1% O_2_) and normoxia (21% O_2_) for 24 h. Relative fluorescence intensity of MMP in L929 (**B**) and HUVECs (**D**). CCCP was the positive control. Scale bar: 100 μm. (**E**) Cells were treated for 24 h and stained with Annexin V/PI. Apoptotic cells were stained with green fluorescence, necrotic cells were stained with green and red fluorescence, and normal cells were not stained. Scale bar: 100 μm. (Data are presented as mean ± S.D. *n = 3*, * indicates significant difference, * *p* < 0.05, ** *p* < 0.01).

**Figure 4 ijms-23-15832-f004:**
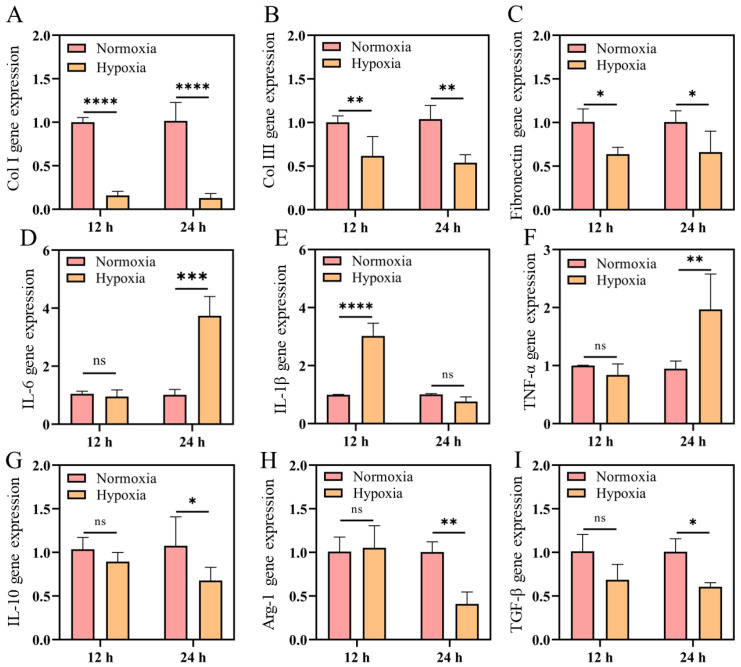
Gene expression of cells cultured under hypoxia (1% O_2_) and normoxia (21% O_2_) for 12 h and 24 h. (**A**–**C**) Gene levels of Col I, Col III and fibronectin of L929. Gene expressions of M1 (IL-1β, IL-6, and TNF-α) markers (**D**–**F**) and M2 (IL-10, arginase-1, and TGF-β) markers (**G**–**I**) of RAW264.7. (Data are presented as mean ± S.D. *n = 3,* n.s. no significance, * indicates significant difference, * *p* < 0.05, ** *p* < 0.01, *** *p* < 0.001, **** *p* < 0.001).

**Figure 5 ijms-23-15832-f005:**
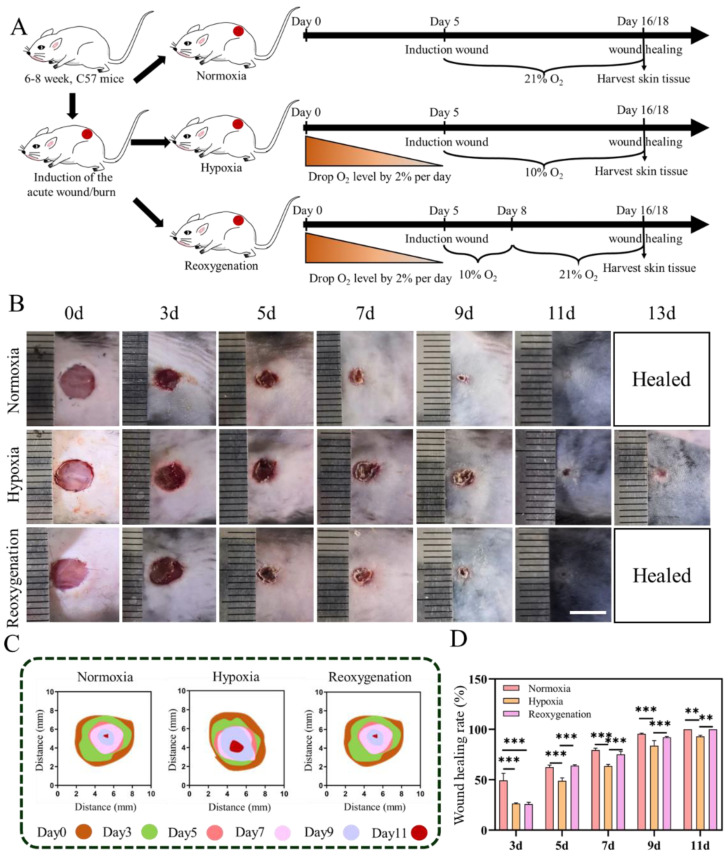
Acute mice wound closure in normoxia, hypoxia and reoxygenation. (**A**) Details of the animal experimental. (**B**) Appearance of wounds at different time. (**C**) Schematic diagram of wounds closure. (**D**) Numerical analysis of wounds healing rate. Scale bar: 5 mm. (Data are presented as mean ± S.D. *n = 6*, * indicates significant difference, ** *p* < 0.01, *** *p* < 0.001).

**Figure 6 ijms-23-15832-f006:**
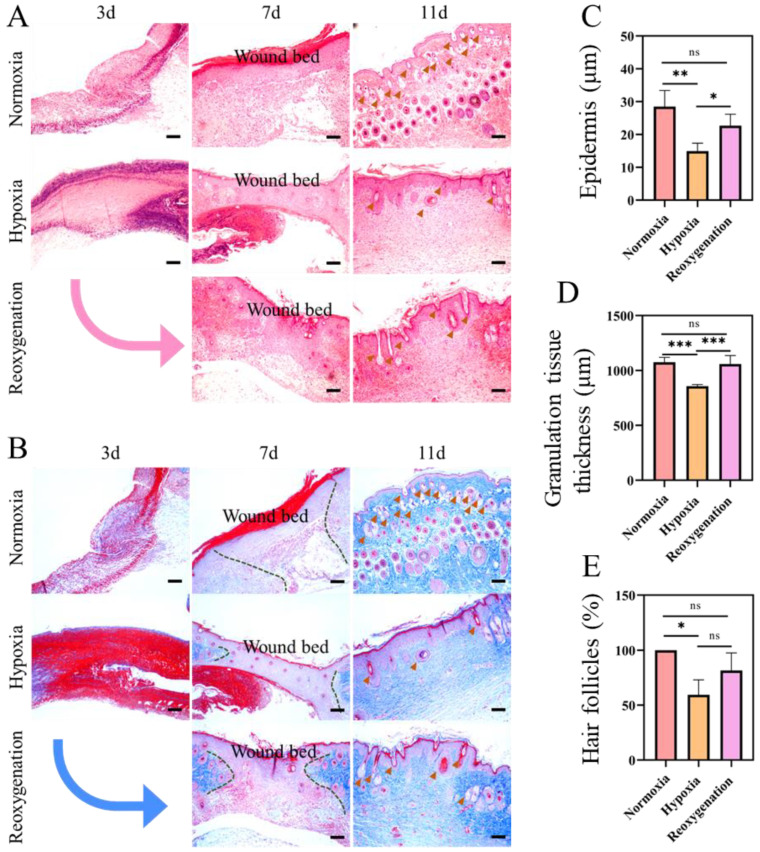
Wound healing and skin regeneration evaluation in mice model at day 3, 7, and 11. (**A**) H&E staining of granulation tissue thickness. (**B**) Masson’s trichrome staining. (The brown arrows are hair follicles. The green dotted line shows granulation tissue). Quantification of the epidermal thickness (**C**), granulation tissue (**D**), and hair follicles (**E**) at day 11 in different groups. Scale bar: 100 μm. The arrow shows the transition from hypoxic to reoxygenated. (Data are presented as mean ± S.D. *n = 6*, n.s. no significance, * indicates significant difference, * *p* < 0.05, ** *p* < 0.01, *** *p* < 0.001).

**Figure 7 ijms-23-15832-f007:**
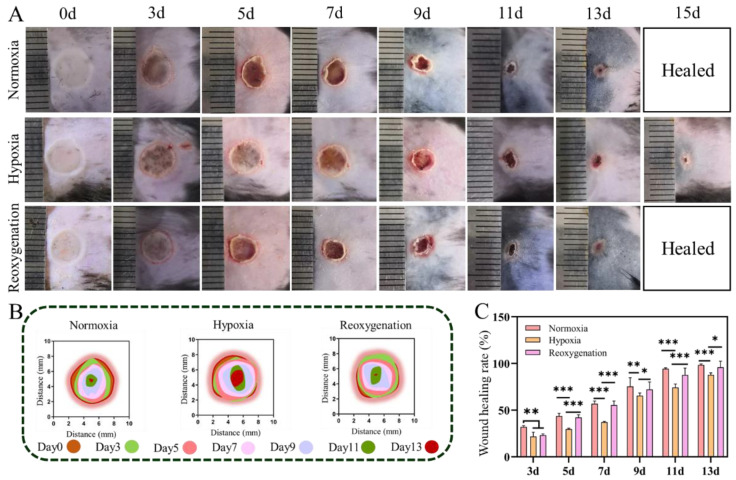
Burn wound closure of mice in different time under normoxia, hypoxia and reoxygenation. (**A**) Representative photographs of burn mice wounds. (**B**) Schematic diagram of the burn wound closure. (**C**) Numerical analysis of wounds healing rate in burn. Scale bar 5: mm. (Data are presented as mean ± S.D. *n = 6*, * indicates significant difference, * *p* < 0.05, ** *p* < 0.01, *** *p* < 0.001).

**Figure 8 ijms-23-15832-f008:**
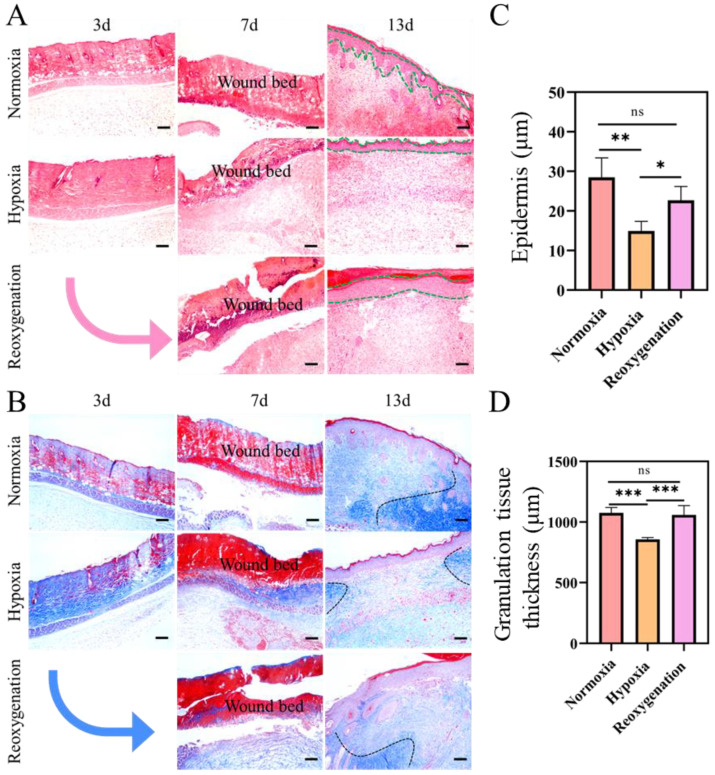
Burn wound healing and skin regeneration evaluation under different oxygen treatment at different time. (**A**) Haematoxylin and eosin staining. The green dotted line represents the epidermis. (**B**) Masson trichrome staining. Quantification of epidermal thickness (**C**), and granulation tissue thickness (**D**) of burn wound tissue on day 13. Scale bar: 100 μm. The arrow shows the transition from hypoxic to reoxygenated. (Data are presented as mean ± S.D. *n = 6*, n.s. no significance, * indicates significant difference, * *p* < 0.05, ** *p* < 0.01, *** *p* < 0.001).

**Figure 9 ijms-23-15832-f009:**
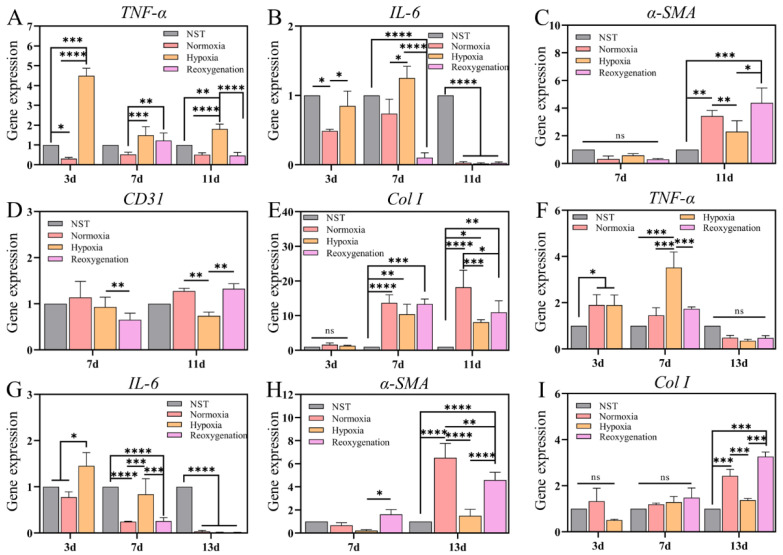
Assay of gene expression of wounds in different time. (**A**,**B**) Inflammatory factor (TNF-α and IL-6) in acute wound on 3, 7, and 11 days. (**C**,**D**) Angiogenic factors (α-SMA and CD31) in acute wound on 7 and 11 days. (**E**) Col I gene expression in acute wound on 3, 7, and 11 days. (**F**,**G**) Inflammatory factor (TNF-α and IL-6) in burn on 3, 7, and 13 days. (**H**) α-SMA gene expression in burn on 7 and 13 days. (**I**) Col I gene expression in burn on 3, 7, and 11 days. All values were against to GAPDH. (NST: Normal skin tissue. Data are presented as mean ± S.D. *n = 3*, n.s. no significance, * indicates significant difference, * *p* < 0.05, ** *p* < 0.01, *** *p* < 0.001, **** *p* < 0.001).

**Figure 10 ijms-23-15832-f010:**
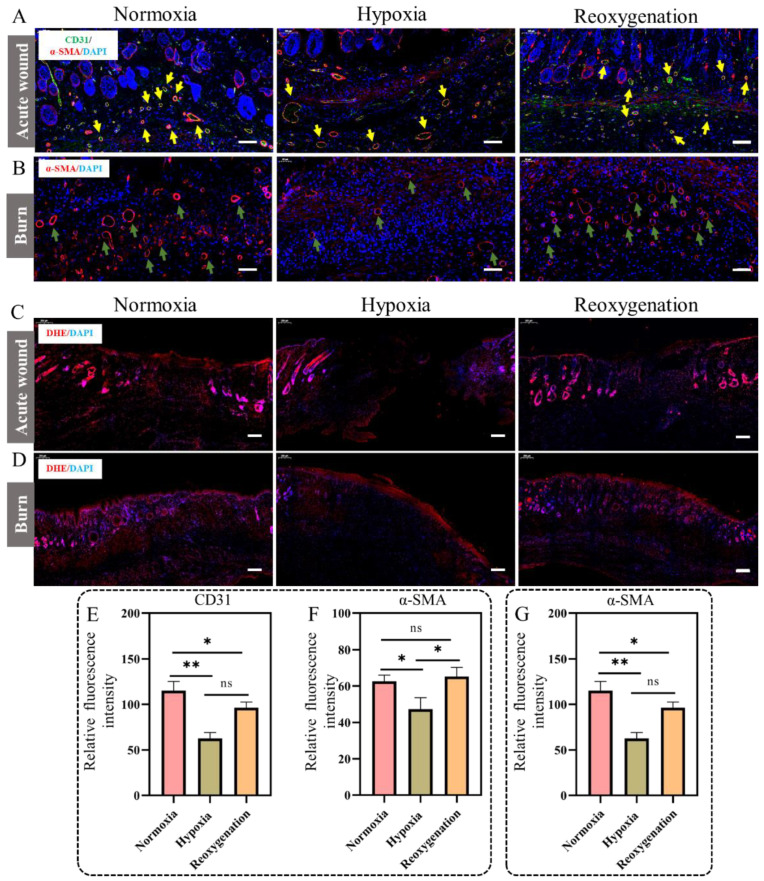
Immunofluorescence assay of wounds after different treatments. (**A**) Immunofluorescence images of CD31 and α-SMA co-expression in acute wounds at 11 days. Blue signal: DAPI; Green signal: CD31; red signal: α-SMA. The yellow arrow represents CD31/α-SMA. Scale bar: 100 μm. (**B**) Immunofluorescence images of α-SMA expression in burn at 13 days. Blue signal: DAPI; red signal: α-SMA. The green arrow represents the α-SMA. Scale bar: 100 μm. The ROS levels in the acute wound (**C**) at 11 days and burn (**D**) at 13 days. Blue signal: DAPI; red signal: ROS. Scale bar: 200 μm. Relative fluorescent intensity of CD31 protein (**E**) and α-SMA (**F**) in acute wounds. (**G**) Relative fluorescent intensity of α-SMA in burns. (Data are presented as mean ± S.D. *n = 3*, n.s. no significance, * indicates significant difference, * *p* < 0.05, ** *p* < 0.01).

**Figure 11 ijms-23-15832-f011:**
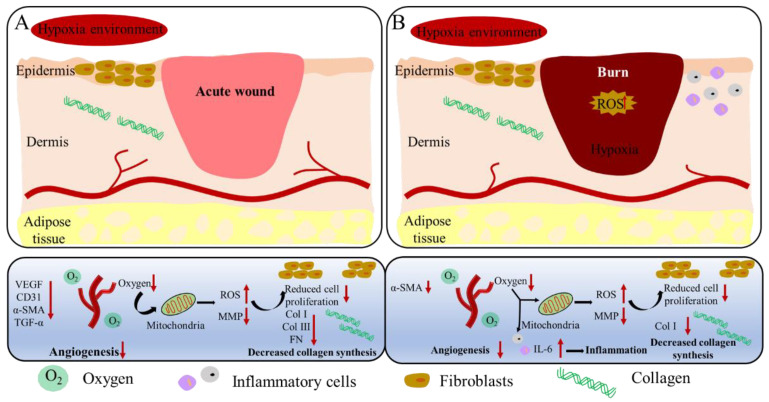
A schematic illustration of the possible mechanisms for delayed healing of acute wounds (**A**) and burn wounds (**B**) under hypoxia.

## Data Availability

Not applicable.

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
