# Peer review of "Reoxygenation Modulates the Adverse Effects of Hypoxia on Wound Repair"

_ijms, 2022, doi:10.3390/ijms232415832_

Round 1
Reviewer 1 Report (New Reviewer)
In this manuscript repair mechanism after hypoxia and reoxygenation in wound healing has been investigated. The design of work is good. So, I recommend it for publication.
Author Response
Thank you for your comments on the manuscript and good luck with your work. We have carefully revised potential errors in the manuscript. Thanks again for your suggestions.
Reviewer 2 Report (New Reviewer)
1- The title is not resembling an original article but seems to be a review. The title should be revised according to the results and conclusions of your work.
2- The mechanisms by which hypoxia exerts its inflammatory conditions should be introduced in the introduction and discussed in the discussion.
3- Figure 1D scale bars are not clear. Live and dead cells should be marked by arrows and if possible figures with higher magnitudes should be provided.
4-Limitations of your work should be better discussed.
Author Response
1- The title is not resembling an original article but seems to be a review. The title should be revised according to the results and conclusions of your work.
Our answer: Thank you for the comments. According to the suggestions of reviewers, we have revised the title of the manuscript. All the revised were marked in red.
Title: Reoxygenation modulates the adverse effects of hypoxia on wound repair
2- The mechanisms by which hypoxia exerts its inflammatory conditions should be introduced in the introduction and discussed in the discussion.
Our answer: Thank you for the useful comments. Introduction: Hypoxia can induce inflammation, and inflamed lesions often aggravate hypoxia, leading to a vicious cycle. Hypoxia is a common feature of the microenvironment of both physiological and pathological immune ecological niches. Exposure to envi-ronmental hypoxia triggers acute inflammatory responses in different organs, including the lungs, the intestine, the kidney, or the heart. The inflammatory response is initiated by the aggregation of leukocytes to the site of infection/injury and this migration process occurs through a series of highly coordinated intercellular interaction steps mediated by cell adhesion molecules and integrins. This increased metabolic activity exceeds the availability of oxygen, leading to inflammation that produces hypoxic conditions locally. Hypoxia at the site of inflammation can lead to vascular rupture, coagulation, ob-struction of blood flow and accumulation of inflammatory cells in hypoxic organs or release of inflammatory mediators (e.g., tumor necrosis factor [TNF], interleukin [IL]-6 or IL-8). Tissue necrosis will occur with the long-term loss of oxygen delivery. Thus understanding the regulation of hypoxia-mediated pathways in inflammation, and the resolution of this process, could provide the basis for tissue damage therapies. (Introduction: Line 71-84)
Discussion and Conclusion: Hypoxia and inflammation are common incidental events in various pathological immune processes, including chronic inflammation and ischemic tissue. The effect of hypoxia on inflammation is mainly driven by HIF-1/2α. HIF-1α and HIF-2α regulate variety of cellular signaling molecules involved in the inflammatory response and can also regulate multi-various immune processes, such as M1/M2 polarization. Hif-1α regulates NF-κB signalling by modulating IκB (NF-κB inhibitor). The interaction between HIFs and NF-κB plays an important role in the hypoxic immune response. Nuclear factor κB (NF-κB) signaling is activated by both inflammatory stimuli (such as cytokines or bacterial products) and hypoxia. Under hypoxic conditions, it was found that increased activity of the inhibitor of κB kinase (IKKβ) and phosphorylation of the κBα inhibitor (IκBα), resulting in the translocation of NF-κB subunits (p65 and p50) to the nucleus, thereby activating transcription of inflammatory genes such as TNF-α, IL-1β, VCAM, and IL-8. In our study, H&E results showed that reoxygenation treatment reduced inflammatory cell infiltration. In addition, qPCR data showed that reoxygenation significantly reduced the expression levels of IL-6 and TNF-α, and effectively alleviated the early inflammatory response of the wound under hypoxia. Therefore, oxygen-carrying materials can be applied in the future to treat inflammatory caused by chronic hypoxic wounds. All the revised were marked in red. (Discussion and Conclusion: Line 450-466)
3- Figure 1D scale bars are not clear. Live and dead cells should be marked by arrows and if possible figures with higher magnitudes should be provided.
Our answer: Thanks for your comments. We increased the size of the scale bars in Figure 1D to make them more legible. Calcein AM staining showed green fluorescence in living cells. While Propidium Iodide (PI) stained dead cells and showed red fluorescence. Dead cells (PI) should be marked by yellow arrows. Furthermore, we re-uploaded high-resolution images and reformatted to make it easier for readers to identify. Thanks again for your suggestions. (Line 126-128)
4-Limitations of your work should be better discussed.
Our answer: Thanks a lot for your comments of our manuscript. By reducing inflammation, increasing angiogenesis, and maintaining collagen deposition, reoxygenation therapy may improve hypoxia-induced chronic wound healing. In the clinical setting, given the pleiotropic effects of reoxygenation therapy, reoxygenation may potentially block the inflammatory cascade and ultimately provide a new therapeutic option for accelerating tissue damage repair. Future studies will involve in-depth characterization of immune cells at the site of inflammation, particularly macrophages. Gene expression profiling techniques such as RNA-Seq are needed to characterize the functional properties of cells at the site of inflammation Further understanding of the mechanisms of reoxygenation for tissue damage is critical for the development of clinical therapeutic interventions, particularly in patients with chronic hypoxic injury. We would like to thank the referee again for taking the time to review our manuscript. All the revised were marked in red. (Line 532-541)
Reviewer 3 Report (New Reviewer)
Overall, I found this manuscript very interesting and congratulate the authors on the volume or work that was undertaken. In general, my main comment is regarding the figure legends. I would advocate that the authors fully explain the results in the legends as they are stand alone. Further, it is not up to the reader to interpret the data.
My more general comments are as follows:
Abstract
-line 15: please replace the word 'analysis' with 'analyse' and delete the word 'of'
Introduction
-line 41/42 sentence beginning with (SBW): 'For example' is grammatically incorrect. Please amend.
-line 44: never use etc in a manuscript. Please delete.
-line 76 (SBW): 'Studies have shown', you state studies, plural, but you have only cited a single reference. Either add more appropriate references or amend the sentence to correctly reflect the change.
-line 78 (SBW): 'However, severe acute' the sentence is grammatically incorrect. Please amend.
-line 82: reactive oxygen species has already been defined on line 68, hence, please use only the abbreviation here.
-line 84: you state many studies but have only cited one reference. Please amend accordingly
-line 88: delete the word 'choosed' and replace with the word 'chosen'
-line 88 (SBW): 'The results showed' and to the end of that paragraph are actually results. I would suggest you delete that section as results do not belong in the introduction.
Results:
-Please see main comments regarding legends above
-line 122: ROS has already been defined (p 68), Please use only the abbreviation and delete ROS written in full.
-line 451: the word 'inflammatory' should be replaced with 'inflammation'
-line 459: please replace the word 'agree' with the word 'agreeance'
-line 462 (SBW): 'ROS is the key' requires referencing. Please amend.
-line 465 (SBW): 'It has been documented' is grammatically incorrect and also requires referencing. Pleas amend.
-line 489 (SBW): 'Previous studies', you state studies but have only cited one reference. Please amend.
-line 548: the word 'in vitro' is Latin and hence, by convention, should be italicised. Please amend.
-line 617: How were the mice euthanised? Please include details in the methods.
Author Response
Our answer: Thanks a lot for your comments of our manuscript. We will be happy to edit the text further, based on helpful comments from the reviewers. And we corrected grammatical errors, deleted redundant sentences, and added references (Line 93-94; Line 477; Line 482) in the manuscript. In addition, we have carefully revised other potential errors in the manuscript. Thanks again for your suggestions. All the revised were marked in red.
-line 617: How were the mice euthanised? Please include details in the methods.
Our answer: Thank you for the useful comments. After 3, 7 and 10 days of treatment, the mice were euthanized by injection of excessive pentobarbital sodium (Xianshiji, China), and the skin tissues were removed for H&E and Masson staining. All the revised were marked in red. (Line 642-643)
This manuscript is a resubmission of an earlier submission. The following is a list of the peer review reports and author responses from that submission.
Round 1
Reviewer 1 Report
The manuscript "The effects of hypoxia and its mechanism of the wounds repair" deals with the investigation of the influence of hypoxia on processes of wound healing. Although the research was described correctly and seems to be correctly done, the overall novelty of the study is not substantial. It is unclear what is the scientific contribution of the study. The abstract does not mention what was done meither which techniques were used. Figures are extremely unreadable, especially fig. 3,5,7 and 11. The text is not written consistently and English language needs improvement and proofreading.
Reviewer 2 Report
Dear Author's
Thank you for being able to see the results of the submitted manuscript.I find the article interesting, although the topic of hypoxia is known,
the information presented may be interesting for researchers dealing with
this issue. I have no critical comments.
best regards
Author Response
Thank you for your comments on the manuscript. We have re-uploaded the manuscript and thank you for your comments.
Reviewer 3 Report
No comments
Author Response
Thank you for your comments on the manuscript and good luck with your work.
Reviewer 4 Report
Bai et al presented a research article showing interesting findings on the effects of hypoxia and reoxygenation on wound healing. Authors utilized various in vitro and in vivo techniques providing compelling evidence of the detrimental impact of hypoxia on tissue repair events such as re-epithelialization and collagen deposition and how reoxygenation can reverse these effects. However, experimental design has some issues, particularly controls of the gene expression analysis and other unclear parts of the results. Authors are kindly encouraged to respond to my comments for this paper to be considered for publication in IJMS.
Title: since the authors included reoxygenation in their experiment, it should be in the title as it is important as hypoxia affects tissue repair.
Introduction:
Adding a small paragraph on wound healing stages and their regulation is recommended
Please mention what has been done so far in the field of hypoxia and tissue repair and cite those papers
some of these papers, for example, include:
https://doi.org/10.1155/2019/2626374
DOI: 10.14348/molcells.2014.0150
Results:
Figure 1.
Figure labels are not clearly showing which color is for normoxia or hypoxia
If viability is shown by percentage, how y-axis shows a value of more than 100? How is the parentage calculated here?
Red PI stain is not evident in the figure. Are dead cells supposed to be stained Red with PI?
Figure 2.
At what time point was ROS level measured? (please explain in the figure caption)
Line 121: furthermore not forthermore
Line 125. I suggest authors perform an annexin V/PI assay to determine if cells were truly experiencing apoptosis/necrosis. Just relying on the MMP assay to deliver this conclusion is not enough.
Figure 3.
is there a way to quantify MMP activity similar to the case of ROS?
What is CCCP? It is not explained anywhere.
N=? in this experiment
Line 138. a reduction in collagen mRNA does not necessarily mean a decrease in synthesis. Authors have to careful about their conclusion. Authors can still do a western plot to confirm these findings.
Line 153,154 Not clear please revise.
Figure 4. Both IL6 and TNFa are up-regulated in normoxia at 24h which contradict the transition toward the claimed resolution of inflammation and activation of M2 macrophages. At the same time. anti-inflammatory cytokines are upregulated at the same time in normoxia which does not make sense
Figure 5.
Figure 5C contradicts Figure5B. 5C is showing wound closed in hypoxia and open in normoxia
Figure 6.
On which day qualification was done? Please mention in the figure legend
Figure 8.
On which day qualification was done? Please mention in the figure legend
Line 279,280. authors have to be careful since inflammation is critical for wound healing as it is responsible for clearing pathogens and tissue debris. Chronic inflammation, however contributes significantly to delayed healing
Figure 9.
a control group showing the basal expression of these genes is critical to accurately assess their changed expression level. the control group is the skin of mice in normal condition (not wounded or burned)
Is there a reason why the authors did not examine pro-inflammatory genes (1L-1b, TNF-a and IL-6) expression in Re-oxygenation
line317. Please explain here how CD31 and alpha-SMA are related to angiogenesis
Figure 10. did authors look at CD31 expression for burn wounds?
Materials:
Please provide an ethics statement
Line 475. what is the culture media used?
Line 503: how the qPCR data was normalized? Was it to non-injured skin tissue (i.e. time 0)
Please provide the accession number for primers
Round 2
Reviewer 1 Report
Dear authors,
thank you for your response. There was no need to explain to reviewers basic mechanisms of wound repair and what hypoxia means to wound healing. This is indeed the major concern with the study, that it does not bring any novelty to the scientific community. Since experiments were carried out correctly and efforts have been invested in the study, we believe that a basic journal would be more suitable for publishing results. Kind regards.
Author Response
Our answer: We will be happy to edit the text further, based on helpful comments from the reviewers. According to the suggestions of reviewers, we have made major revisions to the manuscript, including the Abstract, Discussion and Conclusion. In addition, the role of reoxygenation in the healing of chronic hypoxic wounds is highlighted in the manuscript, which provides a theoretical basis for the treatment of chronic non-healing wounds in the future. All the revised were marked in red. (Line 9-26, and 427-515)
In this study, we investigated whether reoxygenation has a protective effect against hypox-ia-induced acute injury and burn using the C57BL/6 mouse model. C57BL/6 mice were exposed to hypoxia and treated with both acute and burn injuries and were in hypoxia until wound healing. Next C57BL/6 mice were exposed to hypoxia for 3 days and then transferred to normoxic conditions for reoxygenation until wound healing. Finally, skin wound tissue was collected to analysis of healing-related markers such as inflammation, vascularization and collagen. Hypoxia signifi-cantly increased inflammatory cell infiltration and decreased vascular and collagen production, and reoxygenation notably attenuated hypoxia-induced infiltration of inflammatory cells, up-regulation of pro-inflammatory cytokine levels in the wound, and remission of inflammation in the wound. Immunofluorescence analysis showed that reoxygenation increased the expression of the angiogenic factor α-SMA and decreased ROS expression in burn tissues compared to hypoxia-treated animals. Moreover, further analysis by qPCR showed that reoxygenation could alleviate the expression of hypoxic-induced inflammatory markers (IL-6 and TNF), increase angiogenesis (α-SMA) and collagen synthesis (Col І), and thus promote wound healing. It is suggested that oxygen can be further evaluated in combination with oxygen-releasing materials as a supplementary therapy for patients with chronic hypoxic wounds.
Reviewer 4 Report
I thank the authors for their reply. However, I am not satisfied with the changes made and I believe that the manuscript in its current shape is not suitable for publication in this journal. For example, I asked to revise Figure 5C however, the authors were not that careful while revising it and submitted the figure again showing the wrong images where Normoxia wound is more open than hypoxia. Additionally, the authors did not provide the proper controls for gene expression (time 0/non-injured expression value). So I recommend the rejection of this manuscript.
Author Response
We wish to express our sincere thanks for your suggestions and comments, which are very valuable and helpful for revising and improving our manuscript. I am very sorry that the mark of Figure 5c was not changed due to my negligence, only the picture was modified. We have modified and re-uploaded the correct picture in the manuscript. I apologize again for my carelessness.
Additionally, we have performed one additional experiment to re-confirm everything during revision (the proper controls for gene expression,non-injured expression value). We have corrected and marked red in the manuscript. (Line 330-376)